# Utilizing Null Space in Overactuated Systems
## vs
# Avoiding Null Space in Underactuated Systems

Xiangyu Chu

Department of Mechanical and Automation Engineering

The Chinese University of Hong Kong

Email: xychu@mae.cuhk.edu.hk

*Abstract*—In this retrospective, a high-level principle to guide controller design for underactuated systems based on different treatment on null space will be disclosed, which is inspired by [4]. Firstly, we will show what is avoiding null space in underactuated systems. Secondly, we will introduce how this principle evolves. Thirdly, we will show the tricks that were used in the development. Finally, the limitations in [4] and corresponding extensions will be presented.

## I. WHAT IS AVOIDING NULL SPACE?

Over the years, *the principle of utilizing the null space* in overactuated (redundant) systems has inspired and guided researchers to create many *systematic and unified methods* to address multiple task control problems, where the lower task objectives can be projected into the null space of higher task objectives. In underactuated systems, there is no null space that can be found directly due to the absence of redundant Degree of Freedom (DoF). Lacking a *high-level principle* like the null space utilization, many controllers for underactuated systems were complex and non-intuitive, and more often were designed for specific systems or specific class of systems. There are only very few unified robust feedback controllers for a wide range of underactuated systems.

Unlike the built-in null space in redundant systems, the null space in underactauted system is created intentionally by us so that we can create a corresponding high-level principle to control underactuated systems. In this sense, the principle of solving an underactuated control problem becomes more tractable or predictable. Specifically, the null space in underactuated systems occurs when one considers the control problem from the *inverse* perspective. Let us consider a control-affine nonlinear system of the form

$$\dot{\boldsymbol{x}} = \boldsymbol{f}(\boldsymbol{x}) + \sum_{i=1}^{m} \boldsymbol{g}_i(\boldsymbol{x})u_i, \qquad (1)$$

where $\boldsymbol{x} \in \mathbb{R}^n$ is the generalized coordinate vector, $\boldsymbol{g}_1(\boldsymbol{x}), \cdots, \boldsymbol{g}_m(\boldsymbol{x})$ can be expressed as a mapping matrix $\boldsymbol{J}(\boldsymbol{x}) \in \mathbb{R}^{n \times m}$, acting as a kind of Jacobian ($m < n$), and $\boldsymbol{f}(\boldsymbol{x}) \in \mathbb{R}^n$ is the drift vector field. The goal is to design the control $\boldsymbol{u} = [u_1, u_2, \cdots, u_m]^T$ such that the system evolves from any initail state $\boldsymbol{x}_0$ to the desired state $\boldsymbol{x}_d$. It is challenging since $m$ control inputs have to control $n$ state variables, especially for stabilization tasks. For one of the simplest underactuated robots - unicycle, two control inputs, including linear velocity and angular velocity, are supposed to control Cartesian position and orientation. Taking the inverse of (1) mathematically, we have

$$\boldsymbol{u} = \boldsymbol{J}^{+}(\boldsymbol{x})[\dot{\boldsymbol{x}} - \boldsymbol{f}(\boldsymbol{x})], \qquad (2)$$

where $(\bullet)^{+}$ denotes the left pseudoinverse such that $\boldsymbol{J}^{+} = (\boldsymbol{J}^T \boldsymbol{J})^{-1} \boldsymbol{J}^T$. Note that $\boldsymbol{J}^{+} \in \mathbb{R}^{m \times n}$ has a non-trivial null space with the dimension of $(n - m)$ if $\boldsymbol{J}$ is full rank. Such a non-trivial null space may nullify $\boldsymbol{u}$ even though $\dot{\boldsymbol{x}} - \boldsymbol{f}(\boldsymbol{x})$ is non-zero. *An intuitive idea is to prevent the vector $\dot{\boldsymbol{x}} - \boldsymbol{f}(\boldsymbol{x})$ from falling into the null space of $\boldsymbol{J}^{+}$, i.e., avoiding null space, which is the high-level principle for underactuated system control.*

To visualize the null space in underactuated systems, we use a unicycle model without drift to show it. As shown in Fig. 1, the unicycle is expected to stabilize the position $(x, y)$ and the steering angle $(\theta)$ to $[0, 0, \frac{\pi}{3}]^T$ by controlling the linear drive velocity and angular steering velocity. Red circles, red squares, and the blue line denote a family of initial states with the initial steering angle equal to $\frac{\pi}{2}$, the terminal state of each trajectory, and the null space, respectively. Unfortunately, the unicycle mostly converged to the null space (the blue line), instead of the zero equilibrium point.

In [4], we have firstly implemented the idea of avoiding null space in a tailed robot, where the implementation was inspired by the tail movement. Later on, we

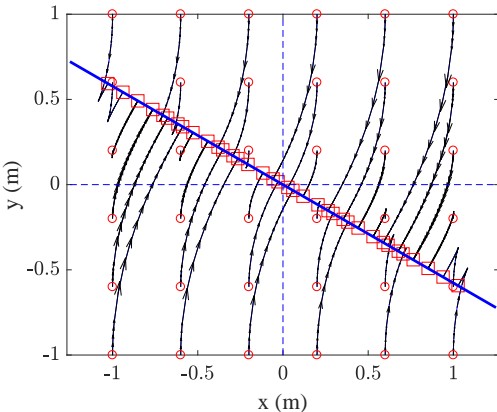

**Figure 1:** Visualization of the one-dimensional manifold or null space at the steady state when solely applying the standard kinematics controllers (such as Jacobian pseudoinverse method) to the unicycle.

generalized this idea to a high-level principle available for multiple underactuated robotic systems (e.g., [5]). To show the effectiveness, we used the generalized controller to solve the trapping problem caused by the non-trivial null space. As shown in Fig. 2, the unicycle can converged to the desired state for all given initial states.

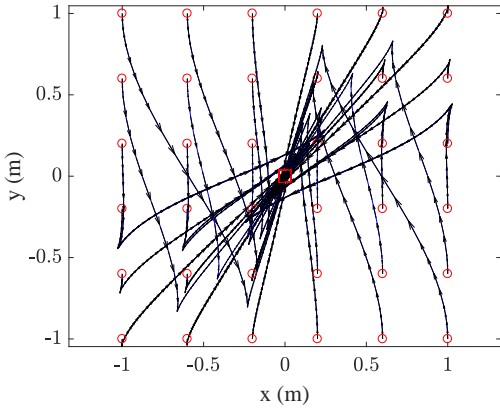

**Figure 2:** Successful stabilization of the unicycle using our generalized controller.

One thing worth mentioning is that the resulting controller for the stabilization control of underactuated systems is almost-continuous, time-invariant and state-feedback. Its discontinuity only occurs at the initial moment. If the system is exactly in the null space, a switching is necessary to help the system escape from the null space. After that, the control will be continuous without any switching. This contribution is not

trivial since there always exists a fundamental obstruction (Brockett's necessary condition [1]) on designing a continuous, time-invariant, and state-feedback control for underactuated systems.

In [2], Brockett also raised a classic question for underactuated system control, "What is the closest one can come to a smooth stabilizing feedback when no smooth feedback control exists?" To our best knowledge, our analytical solution to underactuated control may serve as one of potential solutions to the closest one that can come to a smooth stabilizing feedback when no smooth feedback control exists.

We will omit the details about the implementation of avoiding null space. [4] is a good published paper for illustrating the initial idea of avoiding null space, so the reader is strongly recommended to read Section IV of this paper.

## II. How Avoiding Null Space Evolves?

To our best knowledge, avoiding null space is firstly suggested by Prof. Nakamura when they were trying to plan the motion of underactuated space robots [7]. However, they did not provide any effective controllers and just presented a vision.

[3] gave us lots of inspiration in how to specifically avoid null space. In [3], they used "shape control" (a least squares method) to solve the 3-DoF orientation control by using a 2-DoF actuated tail. Furthermore, they discovered that the tail configuration would affect the error in the "shape control", thus an augmented controller, "perpendicular method", was designed to adjust the tail configuration accordingly to minimize the error.

With the above knowledge, we developed the null-space-avoidance-based algorithm in [4]. Specifically, the "shape control" is the subset of our standard kinematics control; the augmented "perpendicular method" is the subset of our null space avoidance control.

In summary, [4] is the combination of [7] and [3]. [7] provides a guideline of how to solve the stabilization control of underactuated systems, while [3] provides an effective implementation for a specific underactuated system. Proper combination and reasoning allow us to solve the stabilization control in a systematic manner. Currently, the idea of avoiding null space is really implemented in physical robots, rather than only being on the paper.

## III. Tricks

The reader may feel that the control algorithm in [4] is kind of simple. What we want to say is YES! We were challenged a lot since we did not present complex control algorithms to solve challenging underactuated

control problems. The simple but effective controller is beneficial from some tricks.

### A. Selection of $t$ vector

To show our respect to [3], we still use $t$ to represent this key vector in [4]. In [3], $t$ denoted the direction of the tail, which was obtained by observation (see "This vector is the first column of (2) and it can be easily confirmed through inspection" in [3]). Moreover, the $t$ in [3] depended on a simplified model and its existence would be denied if the original system model was considered. Such a $t$ significantly limits the extension to a general case.

Our $t$ vector is more general and can be obtained systematically. It denotes the direction of the null space. More specifically, it is the third left-singular vector of the Jacobian-like matrix if we only consider a 3-state and 2-input underactuated system (see some examples in [4]).

### B. Relation between Cross Product and Skew-symmetric Matrix

Unlike [3], [4] provides a rigorous stability proof. This is attributed to the cross product used in the null space avoidance controller and the conversion between cross product and skew-symmetric matrix. The quadratic form of skew-symmetric matrices is zero, which is a good property while analyzing the derivative of a Lyapunov function.

If the reader pays attention to (25) in [4], you will find that the part of null space avoidance controller is eliminated, as shown in (3)-(9) in this retrospective. One property worth mentioning is $\boldsymbol{J}_{\theta\phi}\boldsymbol{J}_{\theta\phi}^{+}\boldsymbol{S}(\boldsymbol{t}^{M1}) = \boldsymbol{S}(\boldsymbol{t}^{M1})$, where $\boldsymbol{S}(\bullet)$ is the skew-symmetric matrix form of a vector and $\boldsymbol{t}^{M1}$ is the third left-singular vector of $\boldsymbol{J}_{\theta\phi}$. If not $\boldsymbol{t}^{M1}$, this property may not exist.

$$\dot{V} = -\boldsymbol{e}_{\theta}^{T}\boldsymbol{K}^{M1^{T}}\dot{\boldsymbol{\theta}} \tag{3}$$

$$= -\boldsymbol{\lambda}^{M1^{T}}\boldsymbol{J}_{\theta\phi}\dot{\boldsymbol{\phi}} \Leftarrow \text{substituting the kinematics equation and } \boldsymbol{\lambda}^{M1} = \boldsymbol{K}^{M1}\boldsymbol{e}_{\theta} \tag{4}$$

$$= -\boldsymbol{\lambda}^{M1^{T}}\boldsymbol{J}_{\theta\phi}(k^{KM1}\dot{\boldsymbol{\phi}}_{comm}^{K,M1} + k^{NM1}\dot{\boldsymbol{\phi}}_{comm}^{N,M1}) \Leftarrow \text{substituting the proposed controller} \tag{5}$$

$$= -\boldsymbol{\lambda}^{M1^{T}}\boldsymbol{J}_{\theta\phi}\boldsymbol{J}_{\theta\phi}^{+}(k^{KM1}\boldsymbol{\lambda}^{M1} + k^{NM1}\gamma\frac{\boldsymbol{\lambda}^{M1} \times \boldsymbol{t}^{M1}}{||\boldsymbol{\lambda}^{M1} \times \boldsymbol{t}^{M1}||}) \Leftarrow \text{substituting subcontrollers} \tag{6}$$

$$= -\boldsymbol{\lambda}^{M1^{T}}\boldsymbol{J}_{\theta\phi}\boldsymbol{J}_{\theta\phi}^{+}[k^{KM1}\boldsymbol{I}_{3\times3} + k^{NM1'}\boldsymbol{S}(\boldsymbol{t}^{M1})]\boldsymbol{\lambda}^{M1} \Leftarrow \text{rearranging} \tag{7}$$

$$= -k^{KM1}\boldsymbol{\lambda}^{M1^{T}}\boldsymbol{J}_{\theta\phi}\boldsymbol{J}_{\theta\phi}^{+}\boldsymbol{\lambda}^{M1} \Leftarrow \text{using } \boldsymbol{J}_{\theta\phi}\boldsymbol{J}_{\theta\phi}^{+}\boldsymbol{S}(\boldsymbol{t}^{M1}) = \boldsymbol{S}(\boldsymbol{t}^{M1}) \text{ and } \boldsymbol{\lambda}^{M1^{T}}\boldsymbol{S}(\boldsymbol{t}^{M1})\boldsymbol{\lambda}^{M1} = 0 \tag{8}$$

$$\leqslant 0, \tag{9}$$

## IV. LIMITATIONS AND EXTENSIONS

### A. Limitations

Although [4] firstly implemented the principle of avoiding null space, the work in [4] suffers some limitations:

1) Drift Term

   We assumed the initial angular momentum of the tailed robot in flight phase to be zero, thus the discussed system became a driftless system, similar to the unicycle model. However, this case is too ideal in practice and the drift term normally exists.

2) Constrained Input

   Since the method in [4] is a simple feedback control, there is no any constraint involved. In real robots, motors normally show limited capability and the control command may not be satisfied at each moment.

3) Cross Product

   Although cross product helps us design a simple and provable stabilization controller, on the other hand, it limits us to extend the principle of avoiding null space to systems with the dimension of greater than three.

### B. Extensions

To overcome these limitations, we have generalized and extended the work in [4], as follows:

1) Drift Term

   The drift term can be integrated into the standard kinematics part without much modification (the drift term is not limited to be non-zero angular momentum anymore). The rest of design can be similar to that of the driftless case. To show the performance, we take an example of the unicycle-like underwater vehicle and the current plays the role of the drift term. As shown in Fig. 3, the proposed stabilization controller for the drift case can stabilize the vehicle into a bounded area of the

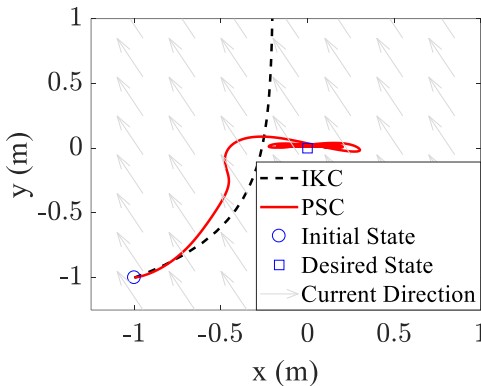

**Figure 3:** 2D trajectories of an underwater vehicle generated by two controllers: Inverse Kinematics Control (IKC) and Proposed Stabilization Control (PSC).

desired state. Note that it is intractable to stabilize the drift system to a desired state precisely and ultimately if the drift is non-vanishing.

2) Constrained Input

Since our method keeps the architecture of the standard kinematics control, we are allowed to integrate it into the Quadratic Programming (QP). Thus, we can include input constraints into the stabilization of underactuated systems. We want to emphasize that the involvement of the QP into the stabilization control of underactuated systems is not trivial and our method is the first solution to our best knowledge. In Fig. 4, we used one example to demonstrate the effectiveness (see black thin lines).

3) Cross Product

[6] has proved that cross product survives not only in 3D space but also in 7D space. Thus, our principle can be applied to a class of 7-state and 6-input underactuated systems (see Fig. 4). Besides, inspired by [8], we directly applied the operation of skew-symmetric matrix into the controller, instead of cross product. This idea facilitates the adaption of our principle to the system not in 3D/7D space, which has been verified in an example of 4-state and 3-input underactuated systems (see Fig. 5).

Our principle is insightful for the control of underactuated systems and shows its capability to a wide range of underactuated systems, such as nonholonomic mobile robots, legged robots in flight phase, underactuated axisymmetric spacecrafts, microswimmers, underactuated ships, and parallel robots. Once having their Jacobian-like mapping matrices, we can intentionally create the null space and apply our principle of avoiding null space

to achieve system stabilization. The principle of avoiding null space opens up a wide range of opportunities for the control of underactuated systems. Besides, the resulting controllers are robust to the drift, which can be rigorously proved. This drift can be interpreted as external disturbance, natural dynamics, and error terms in trajectory tracking tasks.

Overall, we believe that avoiding null space is a fundamental property in underactuated systems and it can guide us to design underactuated system controller systematically. On the one hand, our method can provide lots of underactuated systems with lightweight but effective solutions. On the other hand, the effectiveness of the QP-based methods implies that the principle of avoiding null space can interplay in both analytical and numerical domains. It may be able to be applied to nonlinear optimization like nonlinear Model Predictive Control (MPC) in the sense of cost functions and constraints in the future.

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
