# OpenReview forum: "Utilizing Null Space in Overactuated Systems vs Avoiding Null Space in Underactuated Systems"
_roboticsfoundation.org/RSS/2020/Workshop/RobRetro — RobRetro 2020_

### Official Review · AnonReviewer1 · 2020-06-23
**Interesting method. But as a retrospective it lacks a broader perspective on alternative approaches to the same problem.**

**Confidence:** 4
**Rating:** 5

**Review:**

The paper “Utilizing Null Space in Overactuated Systems vs Avoiding Null Space in Underactuated Systems” proposes a new way to treat the null space in an underactuated system. The authors develop their perspective coherently and explain what they actually consider a null space to be in an underactuated system where the robot has more degrees of freedom than can be controlled.

The problem with this submission is however, that it is difficult to qualify it as a retrospective where the authors should answer: “What should readers of this paper know now, that is not in the original publication?” This could be on Flaws or mistakes in the paper’s methodology, Limitations in the applicability of the work, or Changes in understanding or intuition  (see here for more details https://ml-retrospectives.github.io/how/). On page 4, the authors discuss the limitations of their idea that were brought up in reviews (not general enough) but also discuss some of the remedies. The authors say “t seems reasonable that we have to accept some limitations if we want to enjoy the advantages. ”
However, it is hard to judge the contributions of this paper when there is no quantitative comparison to more general solutions such as inverse dynamics control.

We encourage the authors to conclude such comparison in a retrospective but also original paper (if they haven’t done so - Reference [5] can at the moment not be found online to verify this)

The paper in its current form seems to ask for a broader discussion of what is deemed a contribution in robotics.

---

### Decision · Program_Chairs · 2020-06-25

Accept